# Synthesis of Graphene Oxide from Sugarcane Dry Leaves by Two-Stage Pyrolysis

**DOI:** 10.3390/molecules28083329

**Published:** 2023-04-10

**Authors:** Baskar Thangaraj, Fatima Mumtaz, Yawar Abbas, Dalaver H. Anjum, Pravin Raj Solomon, Jamal Hassan

**Affiliations:** 1Department of Physics, Khalifa University, Abu Dhabi P.O. Box 127788, United Arab Emirates; 2Emirates Nuclear Technology Center, Department of Chemical Engineering, Khalifa University, Abu Dhabi P.O. Box 127788, United Arab Emirates; fatima.mumtaz@ku.ac.ae; 3Molecular Epidemiology and Diagnostic Research Facility, Department of Immunology, School of Biological Sciences, Madurai Kamaraj University, Madurai 625021, Tamil Nadu, India

**Keywords:** biomass, sugarcane dry leaves, two-stage pyrolysis, graphene oxide, two-dimensional

## Abstract

Natural or synthetic graphite as precursors for the preparation of graphene oxide (GO) have constraints due to their limited availability, high reaction temperature for processing of synthetic graphite and higher generation cost. The use of oxidants, long reaction duration, the generation of toxic gases and residues of inorganic salts, the degree of hazard and low yield are some of the disadvantages of the oxidative-exfoliation methods. Under these circumstances, biomass waste usage as a precursor is a viable alternative. The conversion of biomass into GO by the pyrolysis method is ecofriendly with diverse applications, which partially overcomes the waste disposal problem encountered by the existing methods. In this study, graphene oxide (GO) is prepared from dry leaves of sugarcane plant through a two-step pyrolysis method using ferric (III) citrate as a catalyst, followed by treatment with conc. H_2_SO_4_. The synthesized GO is analyzed by UV-Vis., FTIR, XRD, SEM, TEM, EDS and Raman spectroscopy. The synthesized GO has many oxygen-containing functional groups (–OH, C–OH, COOH, C–O). It shows a sheet-like structure with a crystalline size of 10.08 nm. The GO has a graphitic structure due to the Raman shift of G (1339 cm^−1^) and D (1591 cm^−1^) bands. The prepared GO has multilayers due to the ratio of 0.92 between *I_D_* and *I_G_*. The weight ratios between carbon and oxygen are examined by SEM-EDS and TEM-EDS and found to be 3.35 and 38.11. This study reveals that the conversion of sugarcane dry leaves into the high-value-added material GO becomes realistic and feasible and thus reduces the production cost of GO.

## 1. Introduction

Graphene oxide (GO) possesses remarkable physical and chemical properties, enabling its wide usage for technological applications in various fields such as electronics, biomedicine, energy and environment [1,2]. Graphene oxide (GO) is a two-dimensional (2D) carbon sheet consisting of a monolayer of graphite oxide with a polyaromatic structure. It has oxygen-containing functional groups, such as C–O, C=O and –OH, that are anchored on the surface of single-layered graphene. They alter their structure and properties. During the synthesis of GO, the oxidation process creates structural defects that differentiate the physical properties of GO from that of pure graphene. In addition, the surface of GO has many functional groups that exhibit a strong hydrophilic nature. The advantages of GO include its one-atom thickness, good mechanical strength, flexibility, low toxicity and low-cost precursors [3,4,5,6]. In addition, it has a large specific surface area with many hydroxyl and epoxy groups arranged on its basal plane and edges than other inorganic metal oxides and chalcogenide materials [7,8]. 

Chemical routes based on oxidative-exfoliation methods have been developed over the years for the synthesis of GO. These include the methods of Brodie (1859), Staudenmaier (1898), Hofmann (1937), Hummers (1958), modified Hummers—I (1999), modified Hummers—II (2004), modified Hummers—III (2010) and Peng (2015). The use of oxidants, long reaction duration, the generation of toxic gases and residues of inorganic salts, the degree of hazard and low yield are some of the demerits of the above methods [9,10]. Lately, many studies have emerged to modify the above methods, especially Hummers’ method to synthesize GO sheets with a high degree of graphitization [11,12,13,14,15,16]. Generally, GO is synthesized from graphite, but the precursors involved are sparingly available, resulting in high cost. In addition, the processing of synthetic graphite is carried out at extremely high temperatures (≥2500 °C) [17].

Considering the above limitations, biomass wastes offer an alternative raw material for the generation of nanostructured carbon materials. This new approach is economically viable and environmentally safe [18,19,20]. Biomass-derived carbon materials are synthesized by various methods and are widely applied in energy storage, water treatment, catalysis, as adsorbent and in medicine [21,22,23]. In addition, biomass-based carbon materials tend to avoid the growing consumption of renitent chemicals. Recently, a few works have been reported on the synthesis of GO from waste biomass such as coconut shell, rice husk, bagasse [24], tea waste [25], rice straw [7] and oil palm [26]. Pyrolysis technology is the key process in the synthesis of carbon nanomaterials including graphene, carbon nanotubes, carbon fibers, diamond-like carbon coatings and other industrial carbons such as glass-like carbon and graphite [27]. The majority of carbon production from biomass pathways is based on pyrolysis, where biomass precursor is heated to elevated temperatures in an inert gas (Ar or N_2_) atmosphere and in some cases with a catalyst. The selection of biomass and processing conditions have been optimized to yield the nanoscale materials of zero, one, two and three-dimensional carbon structures. Pyrolysis-assisted manufacturing of nanostructured carbon materials from biomass wastes has recently become common. Therefore, this study deals with sugarcane (*Saccharum officinarum*) dry leaf as a precursor for the preparation of GO. This leaf is abundantly available worldwide, and the process demands low temperature and, in a way, negates the involvement of renitent chemicals. Dry leaves of the sugarcane plant are converted into GO via simple two-step pyrolysis using ferric (III) citrate as a catalyst, followed by treatment with conc. H_2_SO_4_. The synthesized GO is analyzed by ultraviolet-visible spectroscopy (UV-vis), Fourier transform infrared spectroscopy (FTIR), X-ray diffraction (XRD), scanning electron microscopy (SEM), transmission electron microscopy (TEM), energy-dispersive X-ray spectroscopy (EDS) and Raman spectroscopy. 

## 2. Results and Discussion

### 2.1. Mechanism of GO Synthesis by Two-Stage Pyrolysis

Many studies hitherto have followed Hummers’ method for the production of GO, considering graphite as starting material. They often used reducing agents such as hydrazine hydrate, sodium borohydride, citric acid, amino acids, and glucose for the reduction of GO. However, they do not efficiently remove oxygen-functional moieties from the GO. Some of the reducing agents are quite toxic and detrimental to living organisms [28]. In contrast, the present study has used sugarcane dry leaves as a carbon precursor instead of graphite and does not deal with any reducing toxic agents. Therefore, it is an environmentally friendly and cost-effective method of synthesis for GO. 

Thermogravimetric (TG) analysis of sugarcane leaves in powder form is performed in the temperature range of ambient to 800 °C with a heating rate of 15 °C min^−1^. Three distinct stages of thermal degradation (weight loss) occur. They are Stage I (ambient–260 °C), Stage II (260–550 °C) and Stage III (550–800 °C). The observed weight loss is only 8 percent in the Stage I due to the removal of moisture and volatile substances [29]. In Stage II, high weight loss (66%) occurs due to thermal cracking of various components of the sugarcane leaves. Complex low volatile organic compounds are generated and released slowly from the leaves in Stage III [30]. The thermal degradation at this stage progressed at a low pace: around 3.3 percent that of Stage II. The residual biomass remains at 22.7 percent at the final stage, revealing the completion of the conversion of biomass into biochar [31,32]. A two-stage heating process is used to carbonize the sugarcane dry leaves under Ar atmosphere. Initially, the mixture containing sugarcane dry leaf and ferric (III) citrate is maintained at 200 °C for 1 h with a heating rate of 3 °C min^−1^. At these conditions, the sugarcane dry leaf is converted into biochar via pre-carbonization. Then, the temperature is raised up to 800 °C with a heating rate of 10 °C min^−1^ and kept at 800 °C for another hour. The synthesis mechanism of GO from sugarcane dry leaves is presented in Figure 1. The components of sugarcane leaves are glucan (33.3%), lignin (36.1%), xylan (18.1%), Arabinan (3.1%), galactan (1.5%) and mannan (1.5%) [33]. During the pyrolysis of sugarcane dry leaves, primary cracking and secondary decomposition lead to the formation of oxygen-functional groups at a temperature range of 400–500 °C [34].

Oxygen is associated with the functional groups of hydroxyl, phenols, ethers, carbonyls and carboxyls in biomass feedstock, while H is associated with the aliphatic compounds and the surfaces of aromatic structures (aromatic C–H). During pyrolysis, the structural H and O are lost while C is condensed into aromatic structures [35]. In addition, it involves covalent bond dissociation of C-heteroatom bonds and rearrangement of the C–C bonds followed by dehydration and elimination of halogens below 500 °C due to their low bond dissociation energies (BDEs) (<450 KJ Mol^−1^). At this stage, rapid weight loss occurs due to the elimination of volatiles [36] and cyclization (formation of aromatic networks) [37]. Bonds with high BDEs (>600 KJ Mol^−1^) are broken and other elements like oxygen and nitrogen are eliminated above 500 °C. During the carbonization process, thermal polymerization takes place and the aromatic networks are interconnected at 800 °C, resulting in primary volume shrinkage and rapid weight loss in the solid [38]. The produced biochar may disintegrate rapidly near the surface of a heated catalyst ferric (III) citrate at 800 °C with a high heating rate of 10 °C min^−1^. Ferric (III) citrate assists in forming graphene-like materials due to the effects of carburization and templating. The Fe^3+^ ions coordinate with the biochar, and iron species diffuse with the carbon atoms of the precursor, to create a dense Fe_3_C layer at 600 °C. At temperatures above >800 °C, the carbon atoms in Fe_3_C release, resulting in the transformation of the material into α-Fe. The out-diffused carbon atoms assembled on the surface of the self-generated Fe template lead to nucleation and subsequent growth of a dense 2D carbon layer like graphene. The high content of Fe from the catalyst creates a layer that can limit the growth of the carbon atoms along the 2D plane due to the template effect [39,40]. The quality of graphene relies on the precursor, inert gas flow rate, catalyst, reaction time and pressure in the furnace, which affects the activation energy required for the formation of graphene nuclei on the catalyst surface [41,42]. The carbonized material of the graphene-like structure is converted into GO by reacting with a strong oxidizing agent (conc. H_2_SO_4_) at 70 °C under air atmosphere. The strong acid reaction damages the surface by breaking the graphitic structure of graphene at 70 °C [43]. Sulfuric acid diffuses between the graphene layers, and quickly reacts with carbon atoms. In addition, it accumulates between the graphene layers, creating randomly oxidized areas all over the flake [44]. Concentrated sulfuric acid is a strong dehydrating agent, which leads to graphene with low oxygen content, fewer defects and high conductivity [45]. There are no reducing agents (HNO_3_, NaOH and KMnO_4_) used in this study. In addition, the strong acid treatment applied here improves graphene oxide by removing metals and other impurities from the surface and increasing the number of acid-functional groups. The treatment also increases the O/C ratio of graphene oxide [46]. Thus, this method is simple and effective for the preparation of GO from biomass wastes. 

### 2.2. Characterization of Synthesized GO

The absorption spectrum of the purified GO is presented in Figure 1. The absorbance peaks of the GO are shown as small and flat bumps ranging from 220 to 320 nm. The band at 220 nm reveals the π-π* electronic transitions of aromatic C=C bonds, indicating the restoration of an extensive conjugated framework of sp^2^ carbon atoms [43]. 

A shoulder peak at 260 nm is caused by the n-π* transitions of aromatic C=O bonds resulting from different oxygen-containing functional groups such as –COOH, –CHO and C–O–C [47,48,49]. An increase in the intensity at 260 nm revealed the presence of a highly oxidized graphene basal plane that forms a large number of isolated aromatic rings or several extended conjugated aromatic rings in the prepared GO, which in turn increases the intensity of C=C bonding peak due to the C=O from the surface of GO [49,50]. 

The presence of different functional groups including oxygen-containing functional groups in GO was identified by the FTIR spectrum (Figure 2). The band at 862 cm^−1^ is attributed to the stretching vibration of C–H while the bands at 1000–1300 cm^−1^ are attributed to the stretching vibration of C–O. The characteristic peaks at 2920 and 2852 cm^−1^ are assigned to the stretching vibration modes of sp^2^ and sp^3^ C–H respectively [51]. The presence of sp^2^ and sp^3^ hybridized carbon atoms confirms that the prepared substance is a two-dimensional (2D) material of GO [52]. The bands at 1022 cm^−1^ revealed the C–O stretching vibrations mixed with C–OH bending from acids, while the band at 1663 cm^−1^ is associated with the C=C skeletal vibrations from aromatic compounds of GO that are easily identified [2,53,54,55]. 

The peak at 1663 cm^−1^ reveals the incomplete oxidation of the graphitic domain [56,57]. The bands at 1900–2500 cm^−1^ are attributed to the stretching vibration of C–H and a broad absorption peak at 3412 cm^−1^ indicating the O–H stretching vibration mode. It is superimposed on the O-H stretch of carboxylic acid due to the presence of absorbed water molecules and alcohol groups. The bands at 1564 and 1152 cm^−1^ correspond to C=C bending and phenolic C–O stretching vibrations. In addition, the deformation of the C–O stretching vibration bands of phenolic and epoxy groups is observed at 1022 and 1152 cm^−1^ [53,54,55,56]. These oxygen-containing groups indicate the formation of GO from graphite-content carbonized material during the oxidation process [53]. These peaks have shown that the presence of such oxygen-containing functional groups significantly increases GO’s hydrophilic properties and interlayer spacing [26].

The XRD pattern of GO is presented in Figure 3. It reveals a dominant peak at 12.47° with an interlayer spacing of 0.71 nm along the plane of (0 0 1), revealing the formation of GO [4,58]. The diffraction angles correspond to the interlayer distance between graphitic sheets [2]. Typically, the interlayer spacing (*d*) of GO ranges from 0.6 to 1.0 nm, which is larger than that of graphite (0.34 nm) [4,58,59]. The increased interlayer spacing in GO depends on the degree of oxidation of graphite and the number of oxygenated functionalities (–COC–, C=O, –OH, COOH) in the basal plane and on the edge of each layer, which enhances the distance between the layers [49,59]. GO is formed by the complete oxidation of graphite, resulting in a broad diffraction peak shift noticed from 26 to 11°. The presence of a broad diffraction peak in the diffraction pattern reveals a loss of coherence between graphene-like layers and very short-range atomic coherence. However, the in-plane peak at 12.47° is sharper, indicating large in-plane structural coherence [60]. There is no sharp peak of graphite occurring at 26°. There is a hump from 21 to 33°, eventually developing a characteristic peak (0 0 2) of reduced graphene oxide (rGO) due to thermal annealing [2,15,61]. The presence of rGO may intercalate with graphene oxide due to the diffraction angle of the rGO/GO occurring at 12.47° [2,16]. In addition, the broad diffraction peak occurred for a high reduction degree of rGO at ~43°, which is associated with the turbostratic band of the disordered carbon materials [56,62].

The crystalline size of the dominant peak of (0 0 1) was calculated using Scherrer’s equation as 10.08 nm. Scherrer’s formula is given below: (1)$$L=K\frac{\lambda }{\beta \;cos\theta }$$
where *L* denotes the crystalline size, *K* denotes the shape factor (0.9), λ denotes the wavelength of the laser source (0.15405 nm), *β* denotes full width at half maximum and *θ* refers diffraction angle between the crystalline planes. 

In general, the Raman spectrum of carbon materials exhibits two major peaks: (i) vibration mode of the disorder (D band) and (ii) vibration mode of sp^2^ carbon atoms (G band). Synthesized GO is presented in Figure 4. The D band appears at 1339 cm^−1^ and is associated with lattice defects from the disordered sp^3^ carbon atoms and the presence of oxygen moieties. On the other hand, the G band at 1591 cm^−1^ is associated with the E_2g_ phonon mode of sp^2^-hybridized carbon atoms in both rings and chains [15,63,64,65,66,67,68]. The intensity of the G band is higher than that of the D band, indicating significant structural disorders due to the harsh oxidation reaction by the acid treatment process. These two bands confirmed that the prepared GO has a graphitic structure. The high disorder in graphite of the prepared GO results in a broader G band and a broad D band with high relative intensity compared with that of the G band [64]. The intensity ratio of *I_D_* and *I_G_* provides an insight into the reduction process by removing oxygen-functional groups that create imperfections within the carbon basal plane and relate to the in-plane sp^2^ crystalline size (*L_a_*) [63]. The crystalline size (*L_a_*) of the materials is inversely proportional to the ratio of *I_D_* and *I_G_*. A low *I_D_*/*I_G_* ratio offers an increase in *L_a_* due to the disordered structure associated with oxygen-containing functional groups [65].

The G band increases dramatically due to the harsh oxidation reaction by the acid treatment, which gives a high *I_D_*/*I_G_* ratio of 0.92. The ratio is consistent with the existing studies [63,64,65,66,67,68,69,70,71]. The 2D band occurs at the high range of 2763 cm^−1^. The importance of the 2D band is to find out the layers of the graphene-like monolayer, double layer and multilayers. Generally, monolayer of graphene occurs at 2679 cm^−1^. The placement of the 2D band at 2763 cm^−1^ confirms that the produced GO is multilayered. In addition, the presence of oxygen-containing moieties inhibits the graphene layer from stacking, resulting in the displaced location of the 2D band [3]. The intensity ratio between the G band and the 2D band is 0.42, which also indicates the presence of multilayers in the synthesized GO. The multilayer of GO has a large interlayer spacing of 0.71 nm due to the dominant peak at 12.47° in the XRD pattern (Figure 3). The formula used for calculating the crystalline size from Raman spectrum is given below:(2)$$L_{a}=\frac{2.4\times 10^{-10}\;\lambda^{4}}{\left(\frac{I_{D}}{I_{G}}\right)}$$
where $L_{a}$ and λ denote the crystalline size of GO and the wavelength of the laser source (532 nm), respectively. The crystalline size for GO was determined as 20.82 nm. It is seen that the difference in crystalline size range is calculated from XRD and Raman spectra. This is due to the fact that crystalline size from Raman spectra depends on the excitation wavelength. A Raman spectrum analyzed at low excitation wavelength is likely to offer crystalline size similar to that obtained from the XRD study [55].

The morphological structure of GO obtained from the dry leaves of sugarcane is shown in Figure 5a.

It is known that the GO has sheet-like structures due to oxygen-containing functional groups incorporated at the edges of the GO [8,16,53,72]. TEM images of GO are shown in Figure 6a. Pristine graphene regions with different degrees of disorder can be observed [73]. It shows a natural tendency to reveal sheet-like and wrinkled edges of transparent carbon sheets. It is also known that several wrinkles are present throughout the sheets. It also consists of many chemically active functional groups like epoxy, carboxyl and hydroxyl on their basal planes [16,53,55,72]. The crystallinity of GO sheets, as studied by the selected area electron diffraction (SAED) technique, indicates the presence of diffused concentric diffraction rings, revealing poor crystallinity of the material, consistent with the XRD data. This study confirms that the prepared GO has a semi-amorphous nature, which is consistent with the existing reports [63,64,65,66,67,68,69,70,71,72]. The elemental compositions of prepared GO have been studied by SEM-EDS and TEM-EDS (Figure 5b and Figure 6b). The ratios of carbon and oxygen are 3.35 and 38.11. 

GO has a number of oxygen-containing functional groups which are covalently bonded to a graphene basal plane and edges with a C/O atomic ratio of 4.46. It creates a strong hydrophilic characteristic that disperses well in water [74,75,76,77,78]. 

The synthesized GO has high carbon (95.68 wt.%), low oxygen (2.51 wt.%) and less than 2 percent of other elements, which reveals high purity. It is important to point out that the presence of sulfur (S) comes from the use of sulfuric acid during the acid treatment process, while other elements are from the sugarcane dry leaves. The sugarcane leaves contain organic matter (25.00%), organic carbon (14.46%), ash content (3.0%), potassium (0.832%), nitrogen (0.683%), calcium (0.497%), magnesium (0.053%), phosphorous (0.049%), total solid (19.07%), total volatile matter (78.5%), C/N ratio (21.27) [79]. In addition, the compositions of biochar from sugar dry leaf are sodium (3.01 wt %), magnesium (5.19 wt %), aluminum (4.56 wt %), silica (8.37 wt %), potassium (5.63 wt %), calcium (10.23 wt %) and manganese (3.13 wt %) [32]. The high purity of the synthesized GO is suitable for various applications such as adsorbent, energy storage, biomedicine and catalysis [9]. A comparative analysis of GO synthesis methods and properties reported in the literature is presented in Appendix A. 

## 3. Materials and Methods

### 3.1. Materials

Ferric (III) citrate (C_6_H_5_FeO_7_) (Sigma-Aldrich, St. Louis, MO, USA), conc. sulfuric acid (H_2_SO_4_, 95%) (VWR Chemicals PROLABO, Lutterworth, UK) and ethanol (C_2_H_5_OH, 95%) (Merck, Rahway, NJ, USA) required for this study were acquired just before the commencement of the project. All the chemicals were used as received without any purification. Dry leaves of sugarcane plants were collected from Khalifa University, AL Arzanah building, San campus, Abu Dhabi, United Arab Emirates.

### 3.2. Preparation of GO from Sugarcane Dry Leaves

The dry leaves of the sugarcane plant were washed thoroughly in tap water and dried under direct sunlight. The leaves were powdered by a grinder and dried in a hot-air oven at 105 °C. A two-step pyrolysis method was followed to carbonize the sugarcane dry leaves. The leaf powder and ferric (III) citrate were mixed well at 2:1 ratio. The combined material was heated at a heating rate of 3 °C min^−1^ to reach 200 °C under Ar atm. in a tubular furnace and it was kept at 200 °C for 1 h. Consequently, the temperature was raised up to 800 °C from 200 °C at a heating rate of 10 °C min^−1^. The material was then kept at 800 °C for another hour. The tubular furnace was then cooled naturally to room temperature. The carbonized material was treated with sulfuric acid (95%) at 70 °C and stirred at 250 rpm for a few hours. Then, it was kept undisturbed at 70 °C for another 2 h. After completion of the reaction, the mixture was diluted with distilled water to reduce the concentration of H_2_SO_4_. The mixture was then centrifuged and the solution was decanted. The resultant material was washed repeatedly with distilled water and later with ethanol several times. The purified material was dried in a hot-air oven at 105 °C to remove the moisture content (Appendix A). The resultant material was characterized by several techniques.

### 3.3. Material Characterization

A UV-Visible spectrometer (Bruker, Billerica, MA, USA), FTIR spectrometer (Bruker IFS 66, Billerica, MA, USA), XRD using Ni-filters Cu Kα radiation (ν = 1.54 Å) at 40 kV and 40 mA (BrukerMeasSrV D2 Phaser, Billerica, MA, USA), Raman spectroscopy (Oxford Instruments, WITec alpha 300 R, Concord, MA, USA), SEM (JEOL, JSM-7610F, Tokyo, Japan), TEM (FEI TECNAI G2 F20, OR, USA) and EDS (Oxford Instruments, INCA-xart, Concord, MA, USA) were employed to characterize the optical properties, morphology, size, structure and composition of the synthesized GO.

A Bruker Lambda 35 UV–Vis absorption spectrometer was used in the range of 200–800 nm with steps of 0.5 nm at room temperature. FTIR spectra with a wave number 400–4000 cm^−1^ were developed using a Bruker IFS 66. The diffraction data of the finely powdered samples were recorded for 2θ angles between 5 and 80°. The Raman spectrum was recorded between 400 and 4000 cm^−1^ on an INCA-xart Raman spectrometer using a 514.5 nm Ar laser at 0.5 mW power. Images were obtained at magnifications ranging from 45× to 30,000×. TEM measurements were carried out in an TECNAI G2 series transmission electron microscope operating at 200 kV.

## 4. Conclusions

GO was prepared from sugarcane dry leaves by a two-step pyrolysis method using ferric citrate as a catalyst, followed by acid treatment of carbonized material with conc. sulfuric acid. Unlike the other methods, no reducing agents such as hydrogen sulfide, hydrazine, sodium borohydride, dimethyl hydrazine, hydroquinone, aluminum powder, citric acid or amino acids were used in this study for the preparation of GO. The synthesized GO had many functional groups such as –OH, C=C, C–O and C–H, with a C/O weight ratio of 3.35, which created strong a hydrophilic nature. In addition, it showed sp^2^ and sp^3^ stretching vibration modes of hybridized carbon atoms at the surface. The dominant peak at 12.47° corresponded to the plane (0 0 1) with a large interlayer spacing of 0.71 nm, suggesting the formation of GO. A hump ranging from 21 to 33° eventually developed a characteristic peak (0 0 2) of reduced graphene oxide (rGO), which intercalated with GO due to the diffraction angle at 12.47°. The D and G bands of the Raman spectrum also revealed the presence of sp^3^ and sp^2^ hybridized carbon atoms. The prepared GO had a number of oxygen-containing functional groups that were covalently bonded to a graphene basal plane and edged with a C/O atomic ratio of 4.46. The morphology showed that the prepared GO had nanosheets with few wrinkled textures on the surface of the GO. This method is simple, rapid, reliable, efficient, cost-effective and environmentally friendly. The synthesis of GO using different biomass wastes with various catalysts and conditions needs to be attempted in the future so as to get a better comprehension of the concept presented here.

## Data Availability

Not applicable.

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
