# Peer review of "Synthesis of Graphene Oxide from Sugarcane Dry Leaves by Two-Stage Pyrolysis"

_molecules, 2023, doi:10.3390/molecules28083329_

Round 1
Reviewer 1 Report
I am sending a review in the attachment.

Author Response
Thank you for reviewing our manuscript. Your comments and suggestions have improved the revised version.
Please see the attachment.

Reviewer 2 Report
After carefully reading, it was found that this article needs major revision because several issues and explanations are still need to be clarified. I recommend it publication in this journal after providing proper improvement in revised version by including suggestion, modification and reply to raised queries which are given below.
1. There are so many biomass materials. Why do you choose sugarcane dry leaves as raw materials? Please give some reasons.
2. Some thermal analysis results of sugarcane leaves, such as TG, are suggested to be added to justify the pyrolysis process analysis in the result and discussion part.
3. To be strict, as synthesized carbonaceous products cannot be named graphene oxide. Because they do not show ultrathin layer structure like graphene oxide.
4. The sample contains many other elements besides carbon and oxygen. The element analyses of sugarcane dry leaves should be provided. Meanwhile, in the scheme1 and Figure 2, no other elements appear. Why do other elements appear in the analyses of EDS?
5. Some photos of GO prepared from sugarcane dry leaves should be added.
6. Why choose 70℃ and concentrated sulfuric acid to prepare GO?
7. What are the variables of this experiment? The sugarcane leaves from different regions and varieties may lead to inconsistency between the results and those in this research. Are the experiments and results reproducible?
8. The properties of GO synthesized by this method should be tested and compared with those prepared by other methods. Are the products different from GO prepared by traditional methods?
9. Specific surface area, pore size distribution, pore volume of as synthesized GO are suggested to be added.
10. Some tests about the purity of GO should be added to indicate the high purity of the sample.
11. There are some grammatical and punctuation errors in this manuscript. The English language should be improved. Tenses are not consistent from sentence to sentence and there are some grammatical errors.
12. Biomass derived carbonaceous products could be synthesized via various methods and widely applied in energy storage devices, waste water treatment, etc. Some typical references are suggested to be cited to enhance the background, e.g. Journal of Bioresources and Bioproducts 2021, 6 (2), 142-151; Journal of Bioresources and Bioproducts 2022, 7 (2), 116-127; Inorganic Chemistry Frontiers 2022, 9, 6108-6123.
Author Response
Thank you for reviewing our manuscript. Your comments and suggestions have improved the revised version.
Please see the attached file.

Round 2
Reviewer 1 Report
The article may be published in its current form.